# Conspicuous corruption: Evidence at a country level

**Panos Louridas**[1]* , **Diomidis Spinellis**[1,2]

**1** Department of Management Science and Technology, Athens University of Economics and Business, Athens, Greece, **2** Department of Software Technology, Delft University of Technology, Delft, The Netherlands

☉ These authors contributed equally to this work.
* louridas@aueb.gr

**Data Availability Statement:** The data that support the findings of this study are owned by the Greek Ministry of Transport. They are available from the ministry under the provisions of law 3979/2011 article 6, but restrictions apply to the availability of these data, which were used under licensefor the

## Abstract

People can exhibit their status by the consumption of particular goods or experiential purchases; this is known as "conspicuous consumption"; the practice is widespread and explains the market characteristics of a whole class of goods, Veblen goods, demand for which increase in tandem with their price. The value of such positional goods lies in their distribution among the population—the rarer they are, the more desirable they become. At the same time, higher income, often associated with higher status, has been studied in its relation to unethical behavior. Here we present research that shows how a particular Veblen good, illicit behavior, and wealth, combine to produce the display of illegality as a status symbol. We gathered evidence at a large, country-level, scale of a particular form of consumption of an illictly acquired good for status purposes. We show that in Greece, a developed middle-income country, where authorities cannot issue custom vanity license plates, people acquire distinguishing plate numbers that act as vanity plate surrogates. We found that such license plates are more common in cars with bigger engines and in luxury brands, and are therefore associated with higher value vehicles. This cannot be explained under the lawful procedures for allocating license plates and must therefore be the result of illegal activities, such as graft. This suggests a pattern of "conspicuous corruption", where individuals break the law and use their gains as status symbols, knowing that the symbols hint at rule-breaking, as long as the unlawful practice cannot be incontestably established.

## Introduction

We will always find ways to exhibit our standing in relation to others; more than a century ago, Thorstein Veblen coined the term "conspicuous consumption" to denote the use of money to obtain goods that display higher social status: "Conspicuous consumption of valuable goods is a means of reputability to the gentleman of leisure" [1]. Such a quest for "reputability" may lead to goods, called *Veblen goods*, demand for which increases on tandem with their prices, because their consumption indicates greatest means of leisure. Veblen goods can be seen as a form of *positional goods*, that is, goods whose value depends on their distribution in the population [2].

current study, and so are not publicly available. The data can be obtained from the Ministry of Infrastructure, Transport and Networks, General Directorate for e-Government, Department for Application Development, Maintenance & Operations. Email: daslef@yme.gov.gr. Data are also available from the authors upon reasonable request and with permission by the Greek Ministry of Transport. The authors have not received any privileged access to the data that other researchers would not have.

**Funding:** The work was partly funded by the Athens University of Economics and Business Research Center, under project code EP-3094-01. The rest of the funding came from the authors' regular salaries as tenured professors at the Athens University of Economics and Business. There was no additional external funding received for this study.

**Competing interests:** The authors have declared that no competing interests exist.

Conspicuous consumption runs deep and predates Veblen. Luxury brands are the archetypical Veblen goods, but conspicuous consumption encompasses art [3], Minoan architecture in Middle Bronze Age Crete [4], consumer products size [5], busyness and lack of leisure time [6] but also, on the opposite side of the spectrum, experiential purchases such as vacations [7], even, arguably, counterculture [8]. Its ubiquity in time and place may be due to its function as part of a signaling system for short-term mating and romance [9, 10].

At the same time, researchers have shown a link between unethical behavior and higher wealth [11] and social class [12], even between illegal behavior and a driver's car cost [13]. More precisely, it appears that higher social class predicts selfish behavior rather than unethical behavior in general, as lower-class individuals are more likely to engage in altruistic unethical behavior [14]. Other work has shown that rich people appear to behave more pro-socially than poor, but that can be explained by the pressures exerted by poverty and the different marginal value of money to rich and poor [15], or that beside narcissism, consientiousness may be associated with unethical behavior [16]. In the more general field of research on dishonest behavior, a lot of work has been carried out, often with contradicting results. A meta-analysis of experimental research on the topic has found that different experimental setups arrive at different conclusions, highlighting the need for further research, with more representative participant pools [17].

In this work we contribute to these inquiries by investigating corrupt behavior at a whole country level using data from vehicle registrations in Greece. We examine whether vehicle owners are involved in rule-breaking in order to acquire a particular form of a positional good: a vanity plate for their vehicle.

Our setting has three distinctive features. First, it is a field study involving data with millions of observations, covering the behavior of the population of a country over a series of years. Second, it examines the creation of a positional good, which, as we will see, is not supposed to exist. Third, the acquisition of this good, a vanity plate, may require some rule-breaking. A vanity plate is a very manifest status symbol, out on the road to see; if some vehicle owners break the rules, they do not shy away from showing it. Perhaps their illicit status is part of their attraction: owners show that they can bend the rules without getting caught.

We suggest naming the phenomenon we describe in this work "conspicuous corruption", for three reasons. First, it entails the use of a positional good as a conspicuous means of displaying status. Second, the acquisition of the good is likely to be the result of corrupt behaviour, both on the part of the buyer and on the part of the seller (who should not exist in the first place). Third, the display of the good is indirectly a display of corrupt behavior, which does not seem to bother the bearer of the good.

Note that our investigations are based on statistical measures. By no means do we prove or suggest that an individual car owner with a distinctive plate has acquired it with corrupt means. Our research only deals with population-wide measures, and it should only be read under this light.

## Materials

In Greece, vehicle plates are composed by a character prefix (currently three characters long) and four numbers, XXX DDDD. Character prefixes are allocated sequentially on a regional basis and are not amenable to preferential treatment: when a vehicle plate is issued, owners have no choice of the prefix. They are also not supposed to have any choice on the number part; numbers are allocated sequentially. Plates are issued to new cars by regional offices of the Ministry of Transport and are not transferable; they are tied to the car license. The application for a new license is often carried out by the car dealership where the sale takes place, as a

service to the buyer. The sequential issuing of license numbers means that, barring geographical differences, there should be no association between number plates and particular vehicles; the owner cannot choose a specific number. (In Athens, which rations road space with an alternate-day travel scheme based on the car plate's last digit, choosing a plate ending in an odd or an even number is an informal but standard practice).

Not all four digit numbers are perceptually equal, however. Some are more distinctive than others; for instance, compare 1111 to 8174. As a result, some number plates are more desirable. Given that the supply of number plates is limitted (by the number of four-digit numbers), there is only a limited supply of distinctive plates. Therefore, although vehicle owners cannot obtain custom vanity plates, they can obtain surrogate vanity plates. To obtain such a vanity plate, baring chance, a car owner must operate illicitly, probably in cooperation with car dealerships and civil servants working at the regional authority departments that are responsible for car registrations. Some have been found to withhold issuing, for a long period, specific license plate numbers [18, p. 181], perhaps to issue them later on to interested parties.

Our data comprise 5 154 762 car registrations, covering all registered cars in Greece up to March 31, 2017. These cars are registered as "private" and their type is recorded as "passenger" or "mixed use"; we have removed vehicle types such as trucks and busses, machinery, etc. As noted above, the salience of a plate is derived only from the number part. We considered the following patterns to be distinctive: xxxx (all same numbers), x000 (thousands), xy00 (hundreds), xxx (three same consecutive numbers), xxyy (two pairs), xyxy (two pairs interlaced), xyyx (palindromes). We treated each pattern as mutually exclusive to all others; for example, the set of x000 plates does not contain the 0000 plates and the set of xy00 plates does not contain the x000 plates. Fig 1 shows a few photos of representative luxury cars with highly distinctive license plates.

Although since 2011 all public sector information in Greece is by default openly available [19], this is more honoured in the breach than in the observance, and implementation of the corresponding law has been patchy [20]. We obtained the data regarding car plate registrations in April 2017 through a Freedom of Information Act (Greek law 3979/2011) request originally submitted in 2013 and, after several followups, resubmitted in March 2017. The anonymized data we obtained contain each car's vehicle plate number (with the initial letters scrambled to maintain anonymity), type, brand, and registration details.

## Results

To measure the distinctiveness of the patterns we need to quantify the surprise upon encountering a vehicle with a particular plate on the street. For this purpose, we used the information content, also known as surprisal, which is equal to the negative base two logarithm of their probabilities [21]. Table 1 shows the patterns, the number of vehicles with plates following each pattern, the information content, the theoretical and the empirical frequencies of each pattern. The theoretical frequencies are calculated from the number of possible patterns appearing in the range 0–9999. Overall, we can see that distinctive numbers are a small percentage of all numbers so that only 5.5% of cars have vanity plates; their information content is considerably bigger than non-vanity plates (4.21 vs. 0.08 bits).

Our data of car registrations do not include the price or the value of the registered vehicle. Moreover, they do not link vehicles with owners, so we are not able to establish directly their social class or wealth. We therefore resort to two proxy measures of the value of the vehicle: first, its engine size and second, whether it belongs to a luxury brand.

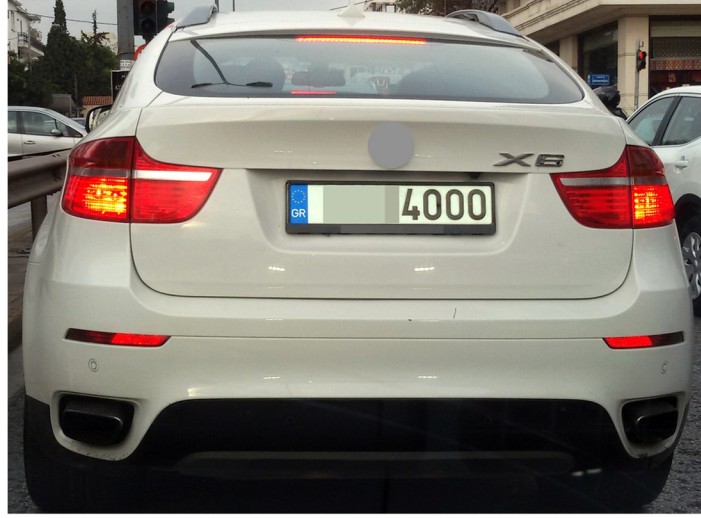
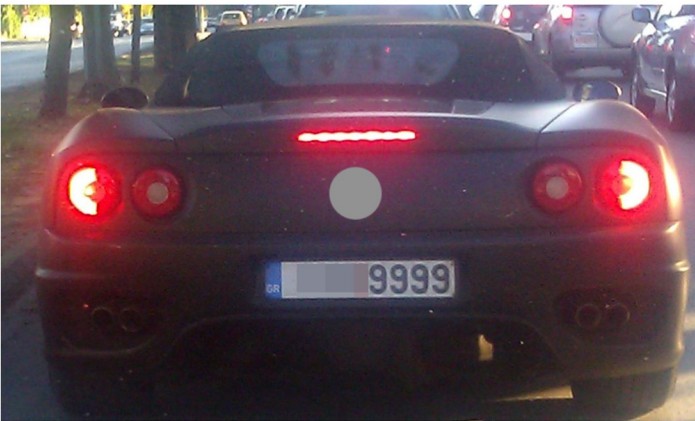
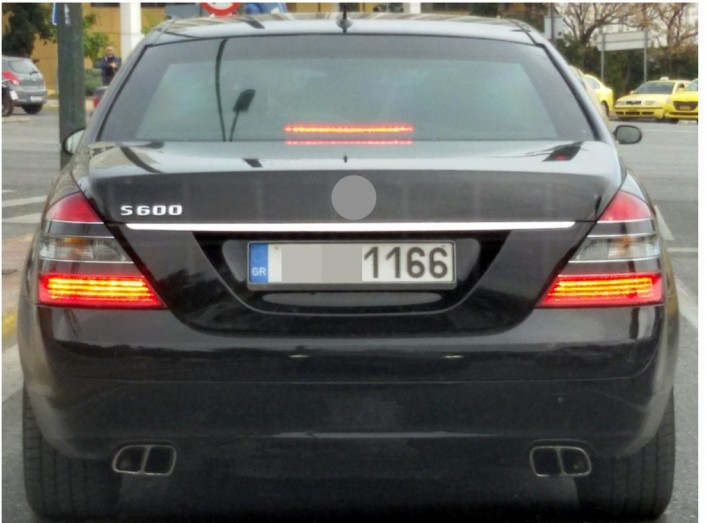
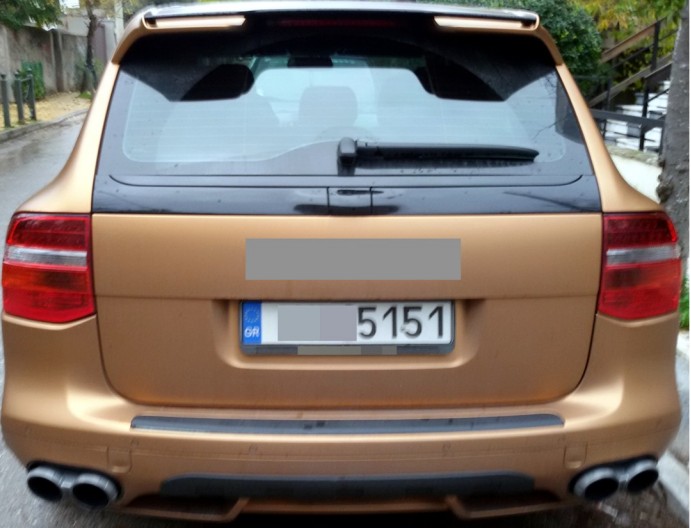

**Fig 1. Examples of luxury car number plate patterns.** Clockwise from top left: x000, xxxx, xyxy, xxyy.

**Table 1. Plate patterns information content and frequencies.**

| Pattern | Count | Information Content | Frequency % | |
|---|---|---|---|---|
| | | | Theoretical | Empirical |
| x000 | 5 581 | 10.12 | 0.09 | 0.11 |
| xxxx | 5 519 | 9.97 | 0.10 | 0.11 |
| xy00 | 48 181 | 7.12 | 0.72 | 0.93 |
| xxyy | 42 598 | 6.80 | 0.90 | 0.83 |
| xyxy | 47 791 | 6.80 | 0.90 | 0.93 |
| xyyx | 47 446 | 6.80 | 0.90 | 0.92 |
| xxx | 88 852 | 5.80 | 1.80 | 1.72 |
| vanity | 285 968 | 4.21 | 5.41 | 5.55 |
| rest | 4 868 794 | 0.08 | 94.59 | 94.45 |
| all | 5 154 762 | | | |

### Vanity plates and engine size

We found that engine size was different for vanity plates compared to non-vanity plates. Our metric for size is engine displacement in cubic centimeters. Overall, cars with any type of vanity plate had a mean engine size of 1586 cm$^3$, while other cars had a mean engine size of 1439 cm$^3$; the difference is statistically significant with the Mann Whittney U test ($p \approx 0$) and the effect size is small to medium; Fig 2 shows a histogram of the two distributions. The effect size varies on the different number patterns. If we break down the data, we see that the effect size increases considerably with more distinctive patterns and becomes very large for the xxxx and x000 patterns [22], presumably the most sought-after, see Table 2 (in all cases $p \approx 0$ for the Mann Whittney U test).

The difference between fancy and plain plates is also shown in Fig 3, which shows the mean engine size of cars for each plate pattern, as well as all of them ("vanity" point) and all non-vanity plates ("plain" point). Blue circles stand for the information content (lower $x$ axis) while

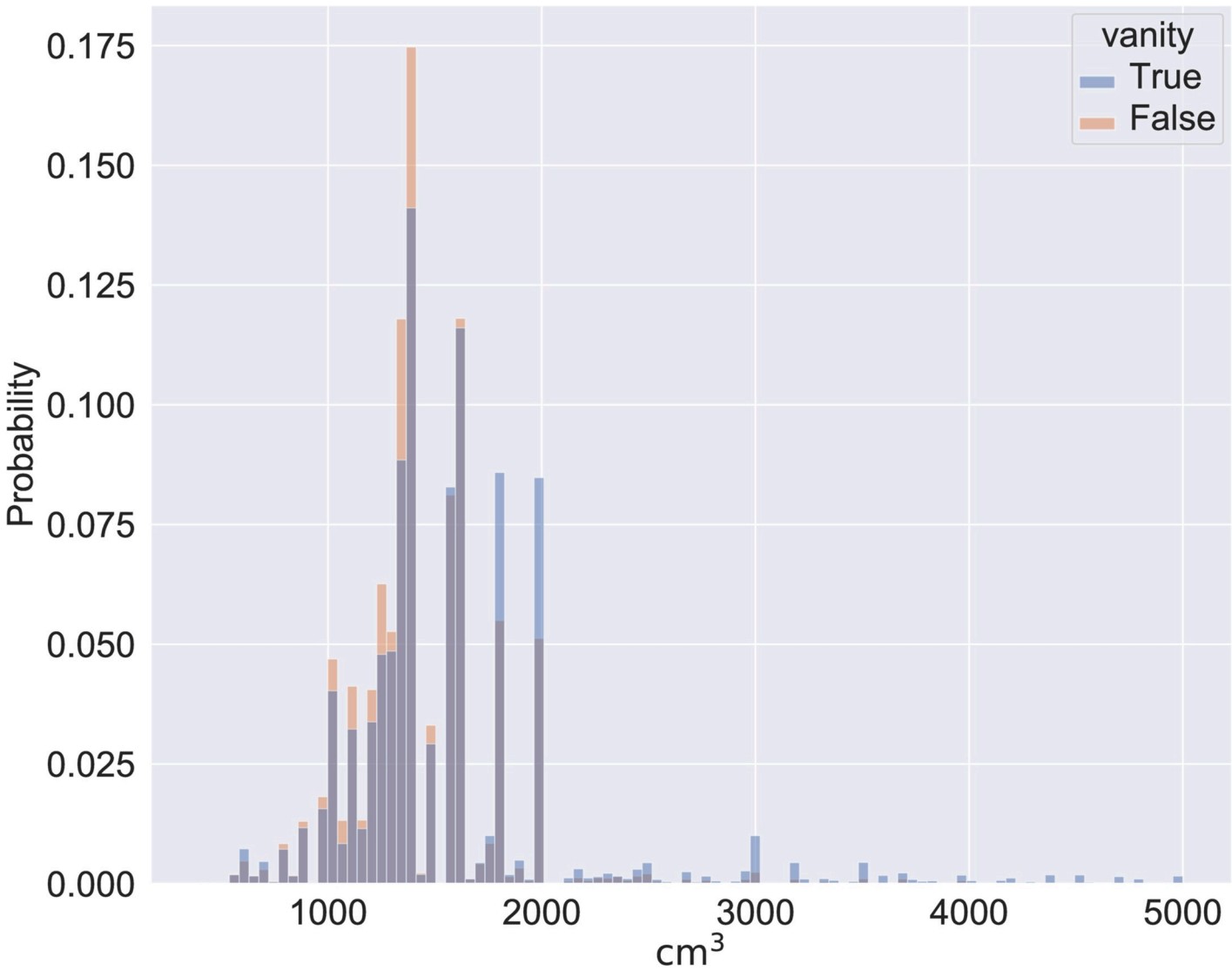

**Fig 2. Histograms of the vanity and non-vanity plates distributions.**

**Table 2. Mean engine sizes, counts and effect size for vehicles with and without specific plate patterns.**

| Pattern | mean | | count | | Cohen's $d$ |
|---|---|---|---|---|---|
| | pattern cm$^3$ | others cm$^3$ | pattern cm$^3$ | others cm$^3$ | |
| xxxx | 1 940 | 1 446 | 5 519 | 5 149 243 | 1.26 |
| x000 | 1 929 | 1 446 | 5 581 | 5 149 181 | 1.23 |
| xyxy | 1 658 | 1 445 | 47 791 | 5 106 971 | 0.54 |
| xy00 | 1 633 | 1 445 | 48 181 | 5 106 581 | 0.48 |
| xxyy | 1 559 | 1 446 | 42 598 | 5 112 164 | 0.29 |
| xyyx | 1 528 | 1 446 | 47 446 | 5 107 316 | 0.21 |
| xxx | 1 521 | 1 445 | 88 852 | 5 065 910 | 0.19 |
| all vanity | 1 586 | 1 439 | 285 968 | 4 868 794 | 0.38 |

white circles stand for the number of vehicles registered under that number pattern (upper $x$ axis). The two values are very close, indicating that surprisal, which is calculated solely based on analytically calculated digit pattern frequencies, is related to the actual, empirically counted, number of such vehicles on the road. The small difference between surprisal and vehicle count for the xy00 and x000 patterns means that these are issued more often than what we would expect.

An engine premium co-occurring with vanity plates should be witnessed also in terms of the probability of vanity plates with regards to engine size. We would expect to see higher probabilities of vanity plates with bigger engines. To investigate that, we counted the number of vanity vs. non-vanity plates for the different engine sizes. However, engine sizes are not determined randomly; for instance, there are about 166 times more cars with engines of 1598 cm$^3$ than with engines of 1600 cm$^3$, because carmakers size their engines taking into account discrete vehicle tax bands. In our data, this results at engine sizes with very few representatives. Among the 10 000 different four-digit numbers, 541 of them correspond to vanity plates. As the probability of a vanity plate is then 541/10000, we need to have at least $10000/541 \approx 19$ cars with a given engine size to expect to encounter a vanity plate. We therefore drop from our counts the engine sizes that are exhibited by fewer cars.

The resulting dataset contains 5 147 252 cars with 879 different engine sizes. The Pearson correlation coefficient between engine size and the probability that a car will have a vanity plate is $\rho = 0.78$ ($p \approx 0$), and the relationship can be approximated through linear regression with $R^2 = 0.60$. If we require more than the minimum cars per engine size and raise the threshold to at least 100, we get 5 130 885 cars with 511 different engine sizes and the correlation improves to $\rho = 0.83$ ($p \approx 0$), while the linear regression improves to $R^2 = 0.69$; see Fig 4. In the latter case, every 1000 cm$^3$ increase in engine displacement is associated with a 6.30 percentage points increase in the probability of the vehicle having a vanity plate. Fig 5 uses the area of markers to indicate the nunber of vehicles with vanity plates for each probability and engine size, for the 100 threshold. For instance, there are 19 527 vanity plates (and 341 092 non-vanity plates) issued to vehicles with engines of 1598 cm$^3$, the most popular engine size.

## Luxury brands and vanity plates

Another proxy indicator of wealth is whether a car is a luxury vehicle. There does not seem to exist a single widely accepted definition of what cars fall into the luxury category, apart from focusing on certain well-known brands [23–27]. Indeed, "The notion of a luxury car varies widely in the literature. For example, Rosecky and King [28] cite five different definitions and

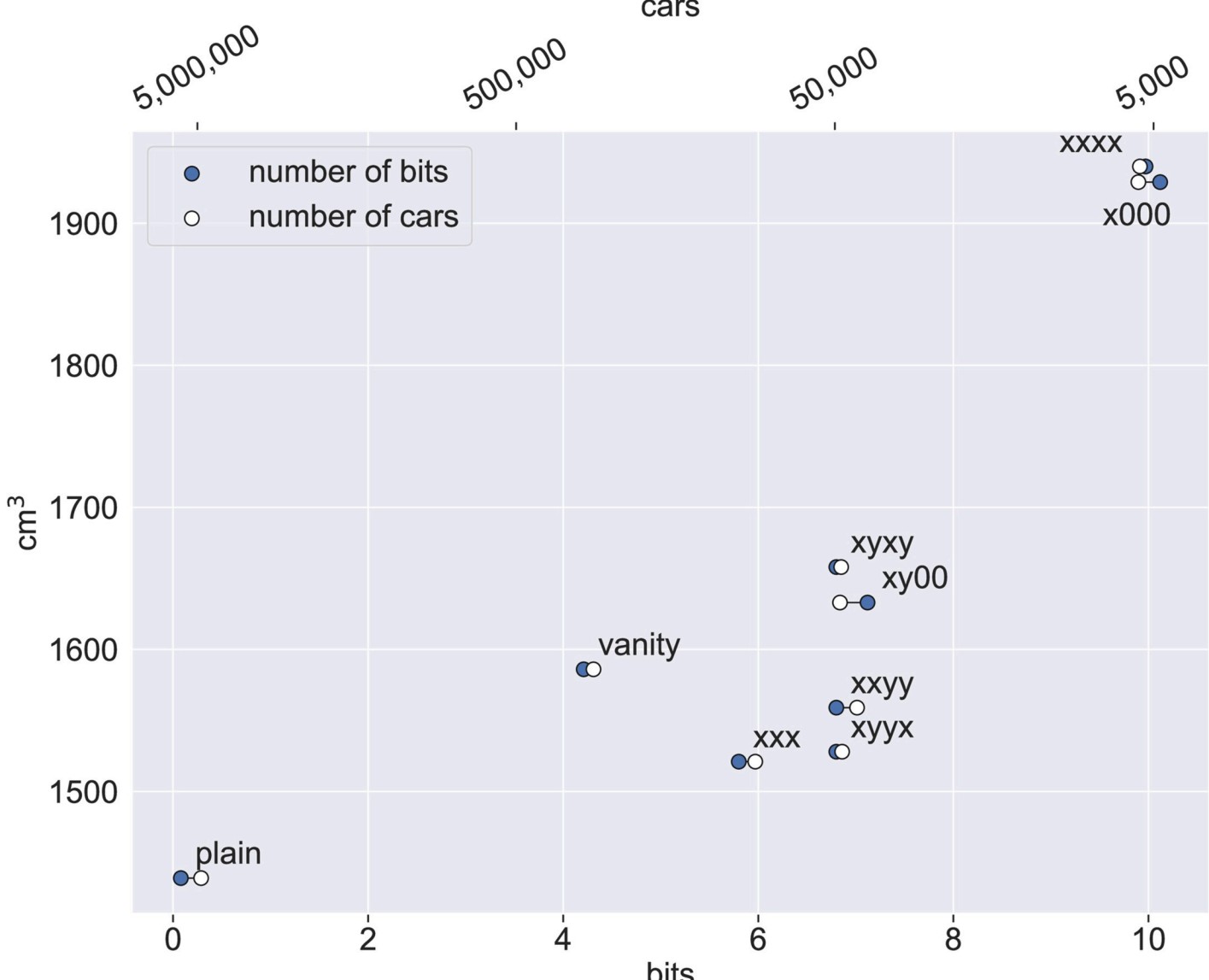

**Fig 3. Cars, patterns, and engine sizes.** For each plate pattern, as well as all vanity patterns and all plain license plates, the mean engine size is on the $y$ axis, the information content of the pattern on the lower $x$ axis and the number of cars on the upper $x$ axis.

limit their study to owners of Mercedes, BMW, Jaguar, Cadillac, Lincoln, Lexus, Infinity, and Acura brands. Is a $50,000 Ford a luxury? What about a $20,000 BMW?" [29].

We proceeded with the Euro Car Segments, used in Europe for categorizing vehicles. In particular, we examined the car sales by category reported by the Hellenic Association of Motor Vehicle Importers-Representatives for the years 2000–2020. We selected those falling under the "F-G" segment, which subsumes the "F: luxury cars" Euro Car Segment. From these we removed brands that produce luxury models but are not mainly or exclusively luxury brands. We found that while the probability of a vanity plate in the whole vehicle population is 5.55%, the probability of a vanity plate on a luxury car is almost double, at 9.88%. Running Fisher's exact test on the vanity vs. luxury car contigency table (see Table 3), we find that that

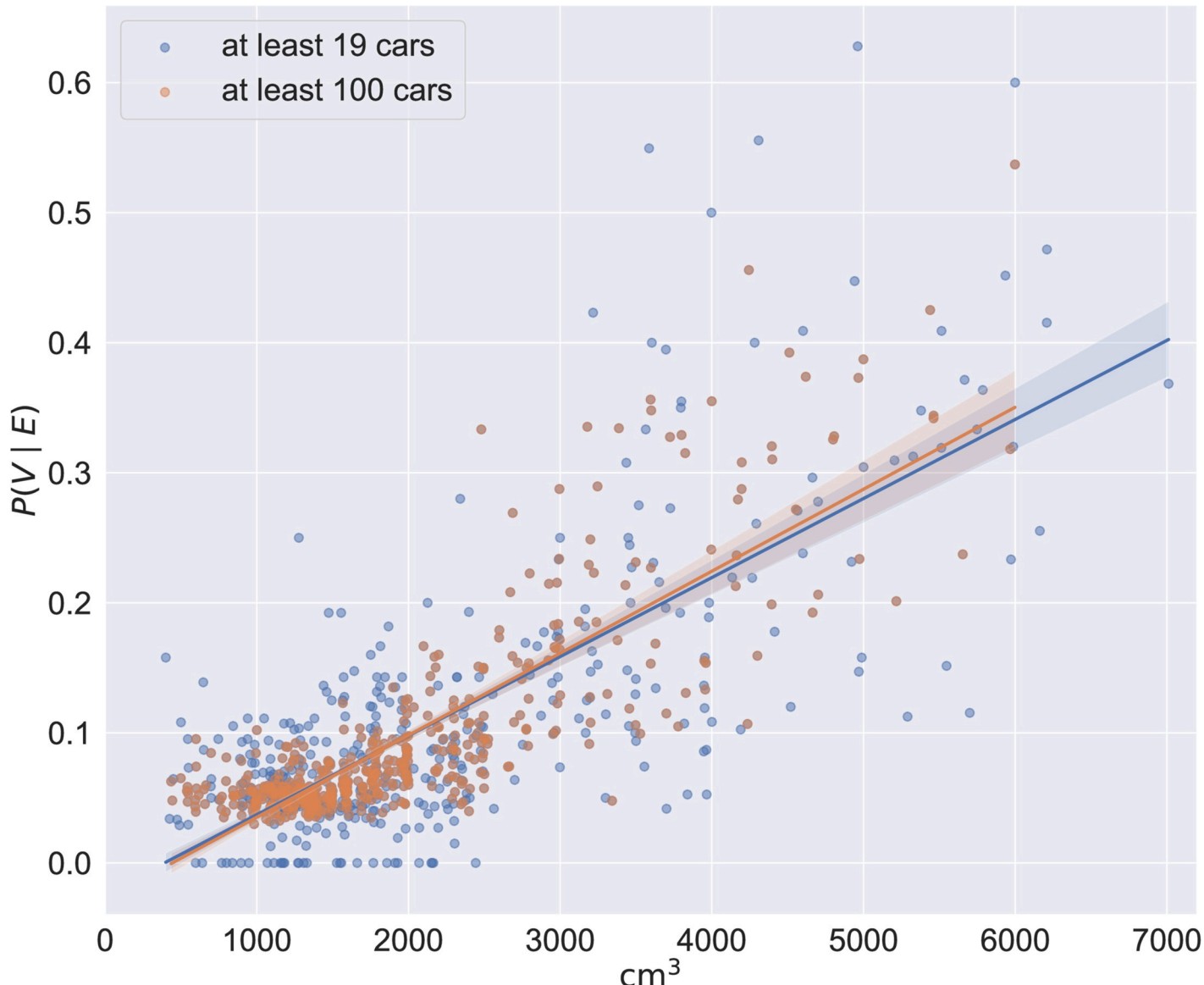

**Fig 4. Probability of vanity plates against engine size.**

we cannot reject the null hypothesis that the proportion of vanity plates is higher among luxury brands than among the rest of the vehicles ($p \approx 0$). We also performed the same analysis basing our luxury criterion on the Wikipedia "Luxury vehicles" category, which is more expansive than the "F-G segment". For instance, the segment does not include Ferrari because it is categorised as a sports, not a luxury car. We arrived at similar results. Apart from the luxury brands, we examined separately Smart cars, a brand made by Daimler AG. Smart have small engines, starting at 599 $cm^3$, and although they are not as expensive as to be luxury models, they have a distinctive design that commands a price premium; therefore they could function as status symbols. We would then expect a high probability of displaying vanity plates. Indeed, we found that the probability of vanity plates among Smart cars is 9.63%.

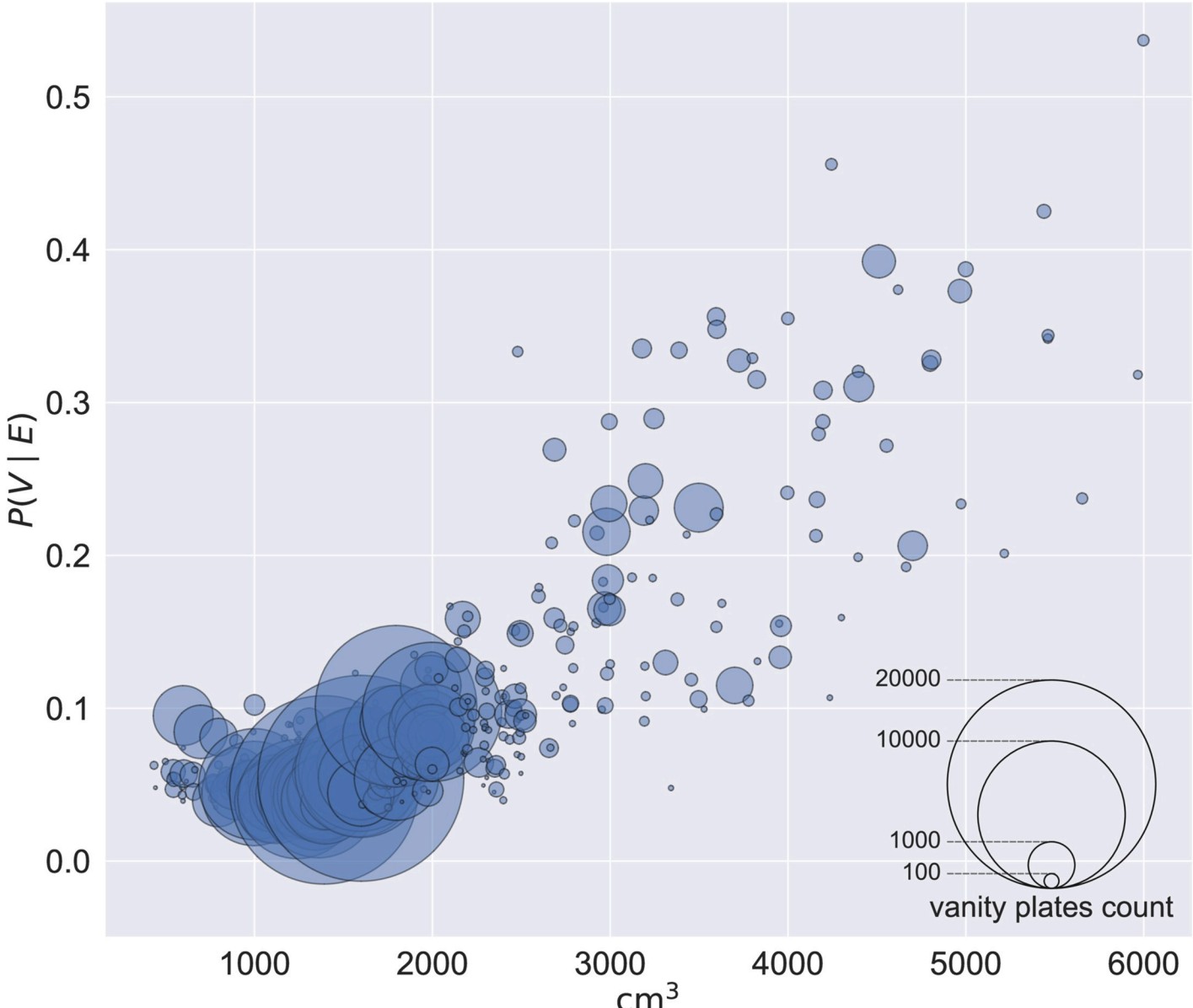

**Fig 5. Probability of vanity plates against engine size; marker area shows number of vehicles.**

**Table 3. Luxury vs. vanity contingency table.**

|            | Luxury  | Non-Luxury | All       |
|------------|---------|------------|-----------|
| Vanity     | 35 397  | 250 571    | 285 968   |
| Non-Vanity | 322 987 | 4 545 807  | 4 868 794 |
| All        | 358 384 | 4 796 378  | 5 154 762 |

## Discussion

The authors embarked on this study spurred by their subjective observations of high-powered cars having disproportionally more distinctive plates than more middle-of-the-road models, bringing forth the suspicion that this could not be random. The data support the suspicion. After some investigation, it appears that the market for vanity plates in Greece is an open secret, the cost for obtaining a desirable number running to a few hundred Euros. Interestingly, rumors have it that during the financial crisis that hit the country in 2009–2018, prices went down, pointing to an elastic market. The situation has not escaped the attention of the authorities. An investigation carried out in 2005 by the inspectors-auditors of the Ministry of Transport "found transgressions in the license registry by withholding special numbers (e.g., 1414, 6666, 8888 etc.) resulting in large gaps in the license numbers registry" (quoted in reference [30]). That means that civil servants in the Ministry of Transport might be withholding the license numbers corresponding to vanity plates, so that they could then be sold. The racket might involve, apart from the buyer, car dealerships, as they are typically the ones going through the process of obtaining the license when a car is bought.

The relatively low price for acquiring a vanity place distinguishes it from classic Veblen goods, which can be much more expensive. Moreover, a vanity plate, even though desirable, is not by itself a mark of wealth: no vanity plate can turn a run-of-the-mill car to a luxury vehicle, even though it has been found in the Netherlands that a particular license plate format, with absolutely no intrinsic value, increased a car's price by about 4% [31].

Once a plate has been issued, there is no way to prove, post fact, that a rule was broken. Also, the violation does not result to immediate societal loss. However, the prevalence of rule violations across societies may impact adversely individual honesty [32]; and a conspicuous disregard of norms questions the merits of following them at all.

Conspicuous corruption is not a form of a criminal status symbol. Criminals use signals to communicate [33], but the bearer of a conspicuous corruption status symbol does not use it to display gang membership or affiliation with a criminal organization. Conspicuous corruption works because it cannot be proved that the law has been broken in a particular instance after the desired number has been issued; a license plate is by itself legal, even though it may have been illegaly acquired.

Concerning the interplay of law and social norms, fighting widespread law breaking is not simply a matter of tightening the screws, if laws contravene social norms: excessively strict or badly designed laws can encourage law-breaking in particular circumstances [34]. Although lawyers may think that laws may shape norms, reality may be different, with groups adhering to their own social norms and contravening the applicable legal rules [35]; the interplay of law and norms is complex [36] and regularization of norms by law is a delicate task [37].

We have studied conspicuous corruption in Greece, which raises the question whether it would arise in different countries. China, in a drive against conspicuous corruption, forbids the use of military plates, which confer formal and informal privileges, on luxury cars [38]. In OECD countries, Greece comes third from the bottom in the average trust of the population in the police; last in the confidence in the national government; and fourth from the bottom in trust in others [39]. It would be interesting to investigate whether conspicuous corruption appears only in low trust societies, or in some other societies with particular cultural characteristics. In terms of corruption in general, although Greece is not a paragon of probity, it is not an egregiously corrupt one. The Control of Corruption estimate in 2019 was at −0.01 in a range of −2.5 (weak) and 2.5 (strong) governance performance [40]. In the Global Corruption Index, Greece is ranked 42 out of 198 countries and territories, achieving a low risk evaluation [41].

Other dimensions that may be pertinent to the prevalence of conspicuous corruption in a country is the degree to which people in the country subscribe to individualism or collectivism and the distribution of power in society (power distance) [42]. Greece, having a culture of intermediate collectivism, is not an outlier; in terms of the power distance it also comes at the middle [43], so it does not appear an extreme case that might limit the findings of this study, at least according to these dimensions.

The question of whether people are more likely to cheat if they are more likely to get away with it, has been studied in the context of reported income for tax purposes. Income underreporting varies significantly across countries, from under 10% to more than 40%, with the share of underreporting not appearing to be related to the development level of the countries, and some of the lowest shares of underreporting occuring in Southern European countries, Bulgaria, Greece, Portugal, and Romania [44]. In the US, on average, the self-employed appear to underreport their income by 25% [45] and the net misreporting percentage (underreported amount divided by the sums that should have been reported) of nonfarm proprietor income (nonfarm businesses owned by a single individual) is at a whopping 56% [46]. A similar pattern has been observed in Denmark, where a study found that while the tax evasion rate was close to zero for income subject to third-party reporting, it was substantial for self-reported income [47]. Based on data gathered from leaks (such as the "Panama papers"), researchers have found that in law-abiding Scandinavia, 0.01% of the richest households, which can avail themselves of offshore heavens, evade about 25% of their taxes [48]. The opportunity to cheat, however, is different from a desire to brag about it; discovering instances of conspicuous corruption in different contexts, in other countries, would shed light on whether we have detected an isolated case or a more widespread phenomenon.

## Conclusions

Through the statistical analysis of Greek car license plates we showed the existence of a particular status symbol whose value as a positional good is bound with the (probably unlawful) process of acquiring it. Expensive cars are by themselves a status symbol. Vanity plates, in jurisdictions where they are lawfully purchased (usually through some bidding process) are an additional status symbol that can be attached to the car. In such cases the value of the symbol is bound with the high price that is assumed the owner has paid for it. The surrogate vanity plates that we have examined are relatively cheap and by themselves alone are not a display of wealth. However, they do bestow some value, otherwise the whole conspicious corruption phenomenon would not exist. For anybody who knows that vanity plates are likely to have been gotten illegally, a luxury car displaying them reveals that the owner "knows the ropes" and that they are probably not particularly disturbed by any potential opprobrium (when legal sanctions cannot be applied); apart from being wealthy, they are also in some place above the rules.

In addition to examining conspicuous corruption, our research demonstrates that it is now possible to study behavioral issues at a big scale, leveraging big amounts of data that are increasingly available and taking a quantitative stance that can complement qualitative studies.

Lastly, conspicuous corruption shows that markets will spring into existence even when they are not supposed, or are prohibited, to be created, if the opportunity exists. One solution could be to combat unlawful behavior; another one would be simply to bring such markets into the fold, accept and regulate them.

## Acknowledgments

We thank Alina Mungiu-Pippidi and Nikos Passas for their advice on corruption metrics.

## Author Contributions

**Conceptualization:** Panos Louridas, Diomidis Spinellis.

**Data curation:** Diomidis Spinellis.

**Investigation:** Panos Louridas, Diomidis Spinellis.

**Methodology:** Panos Louridas, Diomidis Spinellis.

**Software:** Panos Louridas, Diomidis Spinellis.

**Validation:** Panos Louridas, Diomidis Spinellis.

**Visualization:** Panos Louridas, Diomidis Spinellis.

**Writing – original draft:** Panos Louridas, Diomidis Spinellis.

**Writing – review & editing:** Panos Louridas, Diomidis Spinellis.

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
