## [Decision Letter · Decision Letter 0]

29 Mar 2021

PONE-D-21-05961

Conspicuous Corruption: Evidence at a Country Level

PLOS ONE

Dear Dr. Louridas,

Thank you for submitting your manuscript to PLOS ONE. After careful consideration, we feel that it has merit but does not fully meet PLOS ONE’s publication criteria as it currently stands. Therefore, we invite you to submit a revised version of the manuscript that addresses the points raised during the review process.

Please consider all the comments

We look forward to receiving your revised manuscript.

Kind regards,

Ahmed Mancy Mosa, Ph.D.

Academic Editor

PLOS ONE

Journal Requirements:

 'Partly funded by the Athens University of Economics and Business Research Center, under project code EP-3094-01.]'

a. Please provide an amended statement that declares *all* the funding or sources of support (whether external or internal to your organization) received during this study, as detailed online in our guide for authors at http://journals.plos.org/plosone/s/submit-now

Please also include the statement “There was no additional external funding received for this study.” in your updated Funding Statement.

Reviewers' comments:

Reviewer's Responses to Questions

**Comments to the Author**

1. Is the manuscript technically sound, and do the data support the conclusions?

Reviewer #1: Yes

Reviewer #2: Yes

Reviewer #3: Partly

Reviewer #4: Partly

2. Has the statistical analysis been performed appropriately and rigorously? 

Reviewer #1: Yes

Reviewer #2: Yes

Reviewer #3: Yes

Reviewer #4: Yes

3. Have the authors made all data underlying the findings in their manuscript fully available?

Reviewer #1: Yes

Reviewer #2: Yes

Reviewer #3: Yes

Reviewer #4: Yes

4. Is the manuscript presented in an intelligible fashion and written in standard English?

Reviewer #1: Yes

Reviewer #2: Yes

Reviewer #3: Yes

Reviewer #4: Yes

5. Review Comments to the Author

Reviewer #1: The main idea and conclusion is done perfectly and where is the analyze was convenient, keep it up and looking forward for more upgrades and more technical updates for the same topic and it’s a big source for more research about it

Reviewer #2: Review of “Conspicuous Corruption”

Thank you for this really interesting article about “conspicuous corruption” in Greece. I found the data and approach to be novel, and it was an interesting and well-written article. I believe the article should be published as it will be an important contribution to the literature on corruption. However, there are a few areas the authors should consider addressing in order to improve the article and strengthen the claims being made:

1) The authors should make clear straight from the beginning that what they can offer is evidence of an association between corruption and vanity plates, but no clear causal evidence using this data source about so-called “vanity plates” being the result of corrupt behavior. The authors do state this in the end, but it really should also appear up front.

2) The theory is quite thin, and discusses Veblen goods but not the literature on corruption drivers. I’d rather like to see the discussion about corruption that appears on page 6, and in particular the discussion about the author’s concept of “conspicuous corruption”, moved up to appear prior to the methodology and data section.

3) I think the authors need to do a bit more to convince the reader that vanity plates are truly Veblen goods. First, if I understand it correctly, since there are no legal means to obtain vanity plates in Greece, this means that the trade in vanity plates is a black market and vanity plates are illegal goods, not “normal” luxury items. Is it the luxury car or the vanity plate that is the Veblen good, or is it both? Is a vanity plate a Veblen good if it does not also appear on a luxury car? Second, a vanity license plate seems like a rather cheap Veblen good, and without more knowledge about how the Greek population perceives vanity plates it is hard to understand the true value of vanity plates in Greek society. We would really need to know more about the pricing of vanity plates to truly understand them as Veblen goods: what is the going rate for such a plate?

4) The last question I ask under #3 points to a real gap in this article, but one which may be too large to actually address in the article (in which case, perhaps the authors could point to this as a research gap, and/or do more work themselves in future work): this is more understanding of the Greek context and what is considered to be luxury in the context. At a very basic level, it would be useful to know how the licensing process works. Could it be, for instance, that there are legal loopholes that allow, for instance, purchasers of new cars to request any license they want, and that is why we see the association between car type and vanity plates? On a related point: while I like the concept of “conspicuous corruption”, doesn’t this only work if everyone knows that vanity plates are widely understood to be an ill-gotten good? Is it in fact the case that Greek people generally understand vanity plates to be acquired through corrupt means? Also - the authors refer to a Wikipedia list to ascertain what is a luxury vehicle, but isn’t such a list dependent on context? For instance, the author argue that a Smart car would be considered a luxury vehicle in Greece, but this would not necessarily be the case in other places where Smart Cars are more common and even looked down upon. Another issue is understanding the value of vanity plates - does the Greek population generally consider vanity plates to be a luxury good? Third, more qualitative evidence about how the black market in vanity plates in Greece works would be useful to bolster the author’s claims that they are, in fact, considered luxury goods. On page 6, the authors talk about “rumors” regarding the price of vanity plates, but surely they could also conduct some interviews or find newspaper articles to support this claim?

5) Data: what is the timeframe of the data? Details are provided as to when the data was acquired and the number of observations, but not the range of years and geographical locations covered.

6) It would be useful to provide some real-world examples of cars when discussing engine size. What kind of luxury car would be found in the larger vs. smaller engine size categories discussed on page 4? How do we know these larger engine sizes are not trucks, i.e. working vehicles?

Reviewer #3: First, I would like to thank you for the opportunity to review this paper. In general. I believe the authors have investigated a good research topic, entitled, "Conspicuous Corruption: Evidence at a Country Level." The research area lies in my area of interest. I have published articles related to this topic in leading journals. I have enjoyed reading and evaluating this article as it matches my research interests.

This article describes its objective to explore, analyze and evaluate ideas and perceptions of “Conspicuous Corruption”. The study describes that general public with wealth can show their status by consuming a particular product or experience Purchase; this is known as "apparent consumption": this practice is widespread and explained the market characteristics of the entire category of goods, Veblen goods, demand increases with its price. Commodities' value lies in their distribution among populations - the rarer they are, the more desirable they become. At the same time, higher incomes tend to be associated with higher status levels, which has been studied in its relationship to unethical behavior. Here, we introduce research Show how status symbols, immoral behavior, and wealth come together displays illegal as a status symbol. We collect evidence at a large national-level the scale of a particular form of consumption of unlawful benefits for status purposes.

The study has focused on a critical issue of society. However, I have some suggestions for the authors to enhance this work quality. I am recommending your research for publication. It is a good topic; however, you need to work on my suggestions to reach scientific merit. Make changes one by one as suggested.

Title

Revise the title and cover it with a theme reflecting the central idea of the study. Your title does not display the real sense of the main study.

Introduction section

The introduction is not strong. Expand your introduction to about 800 words. Discuss research gaps identified from the literature. In my opinion, here two points could very interesting in this study.

(1) How can social media play a leading role in educating people about this bad habit?

(2) How corporate social responsibility (CSR) can make people sensible to perform their good social duties and contribute to society.

I strongly suggest you build your study with the idea. Your research will become excellent as you have already explored a great idea. The whole world is facing such issues in society; however, it is becoming common in some developing countries. These rich people can spend this wealth for the social good to become immortal. I am suggesting outstanding studies published in leading journals. Please read these studies, improve your introduction, and cite these articles to enhance your work quality.

Su, Z., McDonnell, D., Wen, J., Kozak, M., Abbas, J., Šegalo, S., Li, X., Ahmad, J., Cheshmehzangi, A., Cai, Y., Yang, L., & Xiang, Y.-T. (2021, 2021/01/05). Mental health consequences of COVID-19 media coverage: the need for effective crisis communication practices. Globalization and Health, 17(1), 4. https://doi.org/10.1186/s12992-020-00654-4

Abbas, J., Aman, J., Nurunnabi, M., & Bano, S. (2019). The Impact of Social Media Lon earning Behavior for Sustainable Education: Evidence of Students from Selected Universities in Pakistan. Sustainability, 11(6), 1683. https://www.mdpi.com/2071-1050/11/6/1683

Hussain, T., Abbas, J., Wei, Z., Ahmad, S., Xuehao, B., & Gaoli, Z. (2021). Impact of Urban Village Disamenity on Neighboring Residential Properties: Empirical Evidence from Nanjing through Hedonic Pricing Model Appraisal. Journal of Urban Planning and Development, 147(1), 04020055. doi:10.1061/(asce)up.1943-5444.0000645

Abbas, J., Mahmood, S., Ali, H., Ali Raza, M., Ali, G., Aman, J., . . . Nurunnabi, M. (2019). The Effects of Corporate Social Responsibility Practices and Environmental Factors through a Moderating Role of Social Media Marketing on Sustainable Performance of Business Firms. Sustainability, 11(12), 3434.

Literature section:

The literature section needs improvement. I suggest the authors to look into the suggested studies to improve literature section. Build your idea how innovative strategies can bring change in the government organizations to change public. The authors add the latest citations to the literature and method sections to enhance the study's quality. Cite these studies in the literature to enhance the quality of your work.

Abbas, J. (2021). Crisis management, transnational healthcare challenges and opportunities: The intersection of COVID-19 pandemic and global mental health. Research in Globalization, 100037. https://doi.org/10.1016/j.resglo.2021.100037

Local Burden of Disease, H. I. V. C. (2021, 2021/01/08). Mapping subnational HIV mortality in six Latin American countries with incomplete vital registration systems. BMC Medicine, 19(1), 4. https://doi.org/10.1186/s12916-020-01876-4

Abbas, J., Zhang, Q., Hussain, I., Akram, S., Afaq, A., & Shad, M. A. (2020). Sustainable Innovation in Small Medium Enterprises: The Impact of Knowledge Management on Organizational Innovation through a Mediation Analysis by Using SEM Approach. Sustainability, 12(6). https://doi.org/10.3390/su12062407

Methods and results

I suggest adding demographic table by covering education level, age, income level and regions. See the suggested study and explore if religiosity level can motivate people to contribute to the society. You can add graphical presentation of your findings. See these studies to improve your work and cite them in the methods and results sections.

Abbasi, K. R., Abbas, J., & Tufail, M. (2021, 2021/02/01/). Revisiting electricity consumption, price, and real GDP: A modified sectoral level analysis from Pakistan. Energy Policy, 149, 112087. https://doi.org/10.1016/j.enpol.2020.112087

Aman, J., Abbas, J., Nurunnabi, M., & Bano, S. (2019). The Relationship of Religiosity and Marital Satisfaction: The Role of Religious Commitment and Practices on Marital Satisfaction Among Pakistani Respondents. Behavioral Sciences, 9(3), 30. https://www.mdpi.com/2076-328X/9/3/30

Abbasi, K. R., Hussain, K., Abbas, J., Adedoyin, F. F., Shaikh, P. A., Yousaf, H., & Muhammad, F. (2021). Analyzing the role of industrial sector's electricity consumption, prices, and GDP: A modified empirical evidence from Pakistan [J]. AIMS Energy, 9(1), 29-49. https://doi.org/10.3934/energy.2021003

Discussion section:

Make a separate heading for discussion section, build it on 1000 words, and improve this section. It is short in your study. It should be around one page and a half. Make it strong. See the recommended studies and improve your sections.

Conclusion

Make a separate heading for conclusion and don’t mix it with the discussion section. Your conclusion should be based on minimum 500 words.

Implications

Make a separate heading for implications of your study. I suggest adding implications heading and briefly explain it in this section.

Limitations

Make a proper heading and discuss it adequately.

Highlight the study’s scientific contribution to scientific knowledge. The authors should explain how this study offers useful insights to the researchers of the tourism industry in the discussion section. The English level needs corrections to meet scientific merit for publication. I accept and endorse this manuscript for publication after minor modifications, as suggested.

Reviewer #4: The manuscript addresses a significant issue. The presentations of ideas are demonstrative. However, some comments need to be taken into account:

1- The main point of the manuscript is the relationships between corruption and wealth in Greece. This was apparent from the title. The authors affirmed that when asserted that the study shows “status symbols, unethical behavior, and wealth, combine to produce the display of illegality” (p1). However, the data is confined to one good “car plates”. I do not think it sufficient to have this data as a generalization basis. The manuscript should have gathered and analyzed data of more than one type of accessories to conclude the relationship between illegal behavior and wealth. The other option is to minimize the scope of research (through title and discussion) to have compatibility between title, scope, and data analysis.

2- Corruption, illegality, and unethicality have been used interchangeably throughout the manuscript. Every term has its meaning which may defer from others in the light of the manuscript hypothesis. In this regard, the manuscript uses some references related to unethical behavior with wealth as a support to his propositions about law-breaking activities. I suggest that authors should review the use for more coherence.

3- The manuscript should have elaborate more in the materials section about the regulations of the acquirement of plate numbers. For instance, the legality of ownership transfer of plates, which is available in many countries worldwide, may affect the whole hypothesis of the manuscript.

4- On page 2, the manuscript states that “ we examine whether vehicle owners engage in rule-breaking in order to acquire a particular form of a positional good”. I did not this examination. Moreover, the manuscript states on page 6 “Conspicuous corruption works because it cannot be proved that the law has been broken”.

5- On page 5, the study cites Wikipedia for a piece of information. I do not think this is an authoritative reference.

6- On pages 5-6, the authors state that “ After some investigation, it appears that the market for vanity plates in Greece is an open secret, the cost for obtaining a desirable number running to a few hundred Euros.” These facts should be proved by statistics or citations.

6. PLOS authors have the option to publish the peer review history of their article (what does this mean?). If published, this will include your full peer review and any attached files.

Reviewer #1: **Yes: **Haitham Medhat Aboulilah

Reviewer #2: No

Reviewer #3: No

Reviewer #4: **Yes: **Nabeel Mahdi Althabhawi

---

## [Author Response · Author response to Decision Letter 0]

15 May 2021

We have responded in detail and included all our responses in a separately uploaded file.

---

## [Decision Letter · Decision Letter 1]

7 Jun 2021

PONE-D-21-05961R1

Conspicuous Corruption: Evidence at a Country Level

PLOS ONE

Dear Dr. Louridas,

Thank you for submitting your manuscript to PLOS ONE. After careful consideration, we feel that it has merit but does not fully meet PLOS ONE’s publication criteria as it currently stands. Therefore, we invite you to submit a revised version of the manuscript that addresses the points raised during the review process.

Please consider the comments of Reviewer 4

The comments of reviewer 3: it is not fair that the reviewer request the author to cite his own papers. Therefore, the author can neglect these comments.

We look forward to receiving your revised manuscript.

Kind regards,

Ahmed Mancy Mosa, Ph.D.

Academic Editor

PLOS ONE

Journal Requirements:

Reviewers' comments:

Reviewer's Responses to Questions

**Comments to the Author**

1. If the authors have adequately addressed your comments raised in a previous round of review and you feel that this manuscript is now acceptable for publication, you may indicate that here to bypass the “Comments to the Author” section, enter your conflict of interest statement in the “Confidential to Editor” section, and submit your "Accept" recommendation.

Reviewer #1: (No Response)

Reviewer #2: All comments have been addressed

Reviewer #3: (No Response)

Reviewer #4: (No Response)

2. Is the manuscript technically sound, and do the data support the conclusions?

Reviewer #1: Yes

Reviewer #2: Yes

Reviewer #3: Partly

Reviewer #4: Yes

3. Has the statistical analysis been performed appropriately and rigorously? 

Reviewer #1: Yes

Reviewer #2: Yes

Reviewer #3: No

Reviewer #4: Yes

4. Have the authors made all data underlying the findings in their manuscript fully available?

Reviewer #1: Yes

Reviewer #2: Yes

Reviewer #3: No

Reviewer #4: Yes

5. Is the manuscript presented in an intelligible fashion and written in standard English?

Reviewer #1: Yes

Reviewer #2: Yes

Reviewer #3: No

Reviewer #4: Yes

6. Review Comments to the Author

Reviewer #1: great work. it makes a very insightful reading . the analysis are very well done and the recommendation is apt. a good demonstration of the principles research thanks again and wish to see more upcoming studies relate to the same issue

Reviewer #2: I am satisfied with the author's revisions and responses to the reviewer comments. I have no further comments.

Reviewer #3: The authors have not worked on the suggested points. This manuscript in the revised form does not meet scientific merit.

Reviewer #4: First of all, I would like to thank the authors for their response. However, I still have two minor comments:

1- Regarding comment No 2. I mentioned that “Corruption, illegality, and unethicality have been used interchangeably throughout the manuscript”. The authors responded by stating “ Following your suggestion, we have gone through the manuscript carefully:

• We now use the term \\unethical" only relating to results of work in the literature.

• We use the term \\corruption" only as in \\conspicuous corruption", except our discussion of

corruption in general in the Discussion section (where we start that part of the discussion by “In terms of corruption in general", so the distinction should be clear). I noticed that there is still an overlap in P2 “ researchers have shown a link between unethical or illegal behaviour and higher wealth”

2- Concerning my comments No 5. therein I stated, “ On page 5, the study cites Wikipedia for a piece of information. I do not think this is an authoritative reference”. I am afraid that the authors' justification is not convincing. Multiple scientific works conceptualised “luxury cars”. For instance : MAHMOUDIAN, ALI. "Role of global suppliers in luxury car industry." (2019).

7. PLOS authors have the option to publish the peer review history of their article (what does this mean?). If published, this will include your full peer review and any attached files.

Reviewer #1: **Yes: **Haitham Medhat Abdelaziz Elsayed Aboulilah

Reviewer #2: No

Reviewer #3: No

Reviewer #4: **Yes: **Nabeel Mahdi Althabhawi

---

## [Author Response · Author response to Decision Letter 1]

22 Jul 2021

There were two remaining issues identified by the reviewers; we responded them in this re-submission; please see the "Response to Reviewers" for details.

---

## [Decision Letter · Decision Letter 2]

28 Jul 2021

Conspicuous Corruption: Evidence at a Country Level

PONE-D-21-05961R2

Dear Dr. Louridas,

We’re pleased to inform you that your manuscript has been judged scientifically suitable for publication and will be formally accepted for publication once it meets all outstanding technical requirements.

Kind regards,

Ahmed Mancy Mosa, Ph.D.

Academic Editor

PLOS ONE

Additional Editor Comments (optional):

Reviewers' comments:

Reviewer's Responses to Questions

**Comments to the Author**

1. If the authors have adequately addressed your comments raised in a previous round of review and you feel that this manuscript is now acceptable for publication, you may indicate that here to bypass the “Comments to the Author” section, enter your conflict of interest statement in the “Confidential to Editor” section, and submit your "Accept" recommendation.

Reviewer #4: All comments have been addressed

2. Is the manuscript technically sound, and do the data support the conclusions?

Reviewer #4: Yes

3. Has the statistical analysis been performed appropriately and rigorously? 

Reviewer #4: Yes

4. Have the authors made all data underlying the findings in their manuscript fully available?

Reviewer #4: Yes

5. Is the manuscript presented in an intelligible fashion and written in standard English?

Reviewer #4: Yes

6. Review Comments to the Author

Reviewer #4: I would like to thank authors to their response to the comments. The authors have addressed all comments.

7. PLOS authors have the option to publish the peer review history of their article (what does this mean?). If published, this will include your full peer review and any attached files.

Reviewer #4: No

---

## [Editor Report · Acceptance letter]

10 Aug 2021

PONE-D-21-05961R2 

Conspicuous Corruption: Evidence at a Country Level 

Dear Dr. Louridas:

I'm pleased to inform you that your manuscript has been deemed suitable for publication in PLOS ONE. Congratulations! Your manuscript is now with our production department. 

Kind regards, 

on behalf of

Dr. Ahmed Mancy Mosa 

Academic Editor

PLOS ONE